# Harnessing Uncertainty-aware Bounding Boxes for Unsupervised 3D Object Detection

## Abstract

Unsupervised 3D object detection aims to identify objects of interest from unlabeled raw data, such as LiDAR points. Recent approaches usually adopt pseudo 3D bounding boxes (3D bboxes) from clustering algorithm to initialize the model training. However, pseudo bboxes inevitably contain noise, and such inaccuracies accumulate to the final model, compromising the performance. Therefore, in an attempt to mitigate the negative impact of inaccurate pseudo bboxes, we introduce a new uncertainty-aware framework for unsupervised 3D object detection, dubbed UA3D. In particular, our method consists of two phases: uncertainty estimation and uncertainty regularization. (1) In the uncertainty estimation phase, we incorporate an extra auxiliary detection branch alongside the original primary detector. The prediction disparity between the primary and auxiliary detectors could reflect fine-grained uncertainty at the box coordinate level. (2) Based on the assessed uncertainty, we adaptively adjust the weight of every 3D bbox coordinate via uncertainty regularization, refining the training process on pseudo bboxes. For pseudo bbox coordinate with high uncertainty, we assign a relatively low loss weight. Extensive experiments verify that the proposed method is robust against the noisy pseudo bboxes, yielding substantial improvements on nuScenes and Lyft compared to existing approaches, with increases of +6.9% $AP_{BEV}$ and +2.5% $AP_{3D}$ on nuScenes, and +4.1% $AP_{BEV}$ and +2.0% $AP_{3D}$ on Lyft. The anonymous code and checkpoints are at https://anonymous.4open.science/r/CBC6/.

## 1 Introduction

Unsupervised 3D object detection (Mao et al., 2023; Wang et al., 2023; Ma et al., 2023), given a 3D point cloud, is to identify objects of interest according to the point locations without relying on manual annotations (You et al., 2022; Zhang et al., 2023; Wu et al., 2024; Zhang et al., 2024b), largely saving extra costs and time (Meng et al., 2021). The applications span various domains, including autonomous driving (Grigorescu et al., 2020; Qian et al., 2022; Yurtsever et al., 2020; Zhao et al., 2023), traffic management (Ravish & Swamy, 2021; Milanes et al., 2012), and pedestrian safety (Gandhi & Trivedi, 2007; Gavrila et al., 2004). Existing unsupervised 3D object detection works generally follow a self-paced paradigm (Zhang et al., 2024b), *i.e.*, estimating some initial pseudo boxes and then iteratively updating both the pseudo label sets and the model weights (You et al., 2022; Zhang et al., 2024a). However, we observe that the initial pseudo boxes inevitably contain misalignments (see Fig. 1 (a, b)). The accuracy of the pseudo boxes is significantly affected by the inherent characteristics of LiDAR point clouds, such as point sparsity, object proximity, and unclear boundaries between foreground objects and the background. In particular, large and nearby objects are usually easy to detect, and thus most estimated pseudo bboxes are accurate. In contrast, most small, distant objects with less sensor information pose inaccurate pseudo bboxes at the beginning. Without rectifying such erroneous pseudo bboxes, the wrong predictions can be accumulated, consistently compromising the entire self-paced training process (see Fig. 1 (c)).

To mitigate the adverse impacts of inaccurate pseudo bboxes during iterative updates, we introduce **U**ncertainty-**A**ware bounding boxes for unsupervised **3D** object detection (**UA3D**). As the name implies, we explicitly conduct the uncertainty estimation (Kendall & Gal, 2017; Gawlikowski et al., 2023; Li et al., 2012) for every pseudo bbox quality. The proposed framework consists of two phases: uncertainty estimation and uncertainty regularization. **(1)** In the uncertainty estimation phase, we

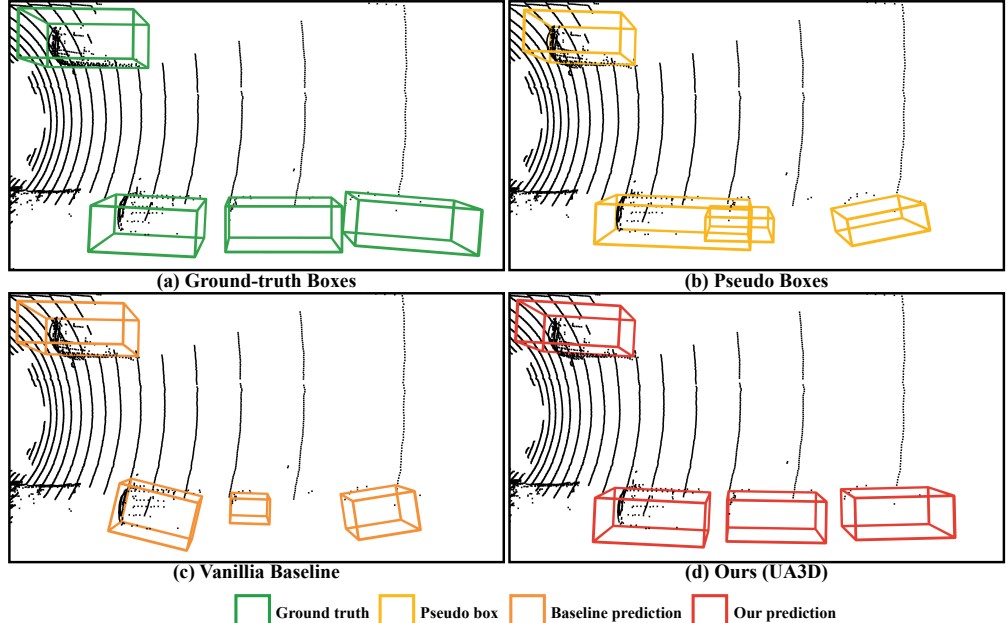

Figure 1: **Our motivation.** Pseudo boxes generated by clustering-based algorithms often contain noise (comparing (a) and (b)). Previous methods (You et al., 2022; Zhang et al., 2023) directly utilize those noisy pseudo boxes to train detection model, leading to suboptimal performance (see (c)). In contrast, we introduce uncertainty-aware pseudo boxes by assigning coordinate-level uncertainty. High uncertainty is assigned to inaccurate coordinates, and during training, the weights of these uncertain coordinates are adaptively reduced. This approach mitigates the negative impact of noisy pseudo boxes, yielding robust detection (comparing (c) and (d)).

introduce an auxiliary branch into the existing detection model, attaching to an intermediate layer of the 3D feature extraction backbone. This branch differs from the original primary detection branch in terms of the number of channels. The uncertainty is assessed by comparing the box predictions from primary and auxiliary detectors. Notably, fine-grained uncertainty estimation on coordinate level is achieved by comparing 7 box coordinates of predictions, *i.e.*, position (x, y, z coordinates), length, width, height, and rotation, from two detectors. **The intuition is that if the pseudo bboxes are with high uncertainty, two detection branches will lead to prediction discrepancy during training procedure.** We could explicitly leverage such discrepancy as the uncertainty indicator. **(2)** In the uncertainty regularization phase, we adjust the loss weights of different pseudo box coordinates based on the estimated uncertainty during iterative training process. Specifically, with the obtained coordinate-level certainty, the sub-loss computed from each box coordinate is divided by its corresponding uncertainty. Meanwhile, to prevent the model from predicting high uncertainty for all samples, the uncertainty value is also added to the sub-loss for each coordinate. This strategy effectively regularizes the iterative training process from noisy pseudo boxes on coordinate level (see Fig. 1 (d)). For example, if a pseudo box is imprecise in its length but accurate in other coordinates, uncertainty is elevated only for length, thereby reducing loss for that specific coordinate. Quantitative experiments on nuScenes (Caesar et al., 2020) and Lyft (Houston et al., 2021) validate effectiveness of our method, which consistently outperforms existing approaches. Qualitative analyses reveal that our model generates robust box estimations and achieves higher recall on challenging samples. Furthermore, uncertainty visualization confirms the correlation between high estimated uncertainty and inaccurate pseudo box coordinates. Our contributions are summarized as follows:

- To mitigate negative effects of inaccurate pseudo boxes for unsupervised 3D object detection, we introduce fine-grained uncertainty estimation to assess the quality of pseudo boxes in a learnable manner. Following this, we leverage the estimated uncertainty to regularize the iterative training process, realizing the coordinate-level adjustment in optimization.

- Quantitative experiments on nuScenes (Caesar et al., 2020) and Lyft (Houston et al., 2021) validate the efficacy of our uncertainty-aware framework, yielding consistent improvements of 6.9% in $AP_{BEV}$ and 2.5% in $AP_{3D}$ on nuScenes, and 4.1% in $AP_{BEV}$ and 2.0% in

AP$_{3D}$ on Lyft, compared with existing methods. Qualitative analysis further verifies that our uncertainty estimation successfully identifies inaccuracies in pseudo bounding boxes.

## 2 RELATED WORKS

**Unsupervised 3D object detection.** Unsupervised 3D object detection endeavors to identify objects without any annotations (Lentsch et al., 2024; Wu et al., 2024; Yin et al., 2022; Luo et al., 2023). This field is distinguished by two primary research trajectories. The first trajectory focuses on object discovery from LiDAR point clouds. MODEST (You et al., 2022) pioneers the use of multi-traversal method to generate pseudo boxes for moving objects, complemented by a self-training mechanism. OYSTER (Zhang et al., 2023) builds on this approach by advocating for learning in a near-to-far fashion. More recently, CPD (Wu et al., 2024) enhances this methodology by employing precise prototypes for various object classes to boost detection accuracy. Additionally, Najibi et al. (2022) employs scene flow to capture motion information for each LiDAR point and applies clustering techniques to distinguish objects. The second trajectory involves harnessing knowledge from 2D space. Najibi et al. (2023) aligns 3D point features with text features of 2D vision language models, enabling the segmentation of related points and bounding box fitting based on specified text, such as object class names. Concurrently, Yao et al. (2022) proposes the alignment of concept features from 3D point clouds with semantic data from 2D images, facilitating various downstream 3D tasks, including detection. Taking one step further, Zhang et al. (2024b) fuses the LiDAR and 2D knowledge to facilitate discovering the far and small objects within a self-paced learning pipeline. Owning to the inherent noise in the generated pseudo boxes, the final efficacy of these approaches can be compromised. Different from existing works, we utilize fine-grained uncertainty estimation and regularization to mitigate the negative effect of inaccurate pseudo boxes to enhance the performance of unsupervised 3D object detection.

**Uncertainty learning.** Uncertainty learning techniques (Xiong et al., 2024; Jain et al., 2024) are broadly categorized into four groups: single deterministic methods, bayesian methods, ensemble methods, and test-time augmentation methods (Gawlikowski et al., 2023; He et al., 2024; Zhang et al., 2024c). Single deterministic methods (Sensoy et al., 2018; Nandy et al., 2020; Raghu et al., 2019; Lee & AlRegib, 2020) adapt the original model to directly estimate prediction uncertainty, though the extra uncertainty estimation usually compromises the original task. Bayesian methods (Neal, 2012; Mobiny et al., 2021; Ma et al., 2015; Wenzel et al., 2020) utilize probabilistic neural networks to estimate uncertainty by assessing the variance across multiple forward passes of the same input, which are limited by high computational costs. Ensemble methods (Sagi & Rokach, 2018; Zheng & Yang, 2021; Ovadia et al., 2019; Malinin et al., 2019; Lakshminarayanan et al., 2017) estimate uncertainty through the combined outputs of various deterministic models during inference, aiming primarily to enhance prediction accuracy, though their potential in uncertainty quantification remains largely untapped. Test-time augmentation methods (Shanmugam et al., 2021; Lyzhov et al., 2020; Magalhães & Bernardino, 2023; Conde et al., 2023) create multiple predictions by augmenting input samples during testing, with the principal challenge being the selection of appropriate augmentation techniques that effectively capture uncertainty. Different from existing techniques, we devise an auxiliary detection branch alongside the primary detector to enable the quantification of fine-grained uncertainty. We also explore the utilization of uncertainty estimation and regularization in the untapped unsupervised 3D object detection task.

**3D object detection framework.** Various 3D object detection frameworks are proposed and operated within a supervised pipeline. Recent works in this domain can primarily be divided into three categories based on the representation strategies: (1) voxel-based, (2) point-based, and (3) voxel-point based approaches. First, voxel-based methods (Zhou & Tuzel, 2018; Yan et al., 2018) transform unordered point clouds into compact 2D or 3D grids, subsequently compressing them into a bird's-eye view (BEV) 2D representation for efficient CNN operations. These approaches, therefore, are generally more computationally efficient and hardware-friendly but sacrifice fine-grained details due to the coarse-grained voxel. Second, point-based approaches utilize permutation-invariant operations to directly process the original geometry of raw point clouds (Shi et al., 2019; Yang et al., 2020; Shi & Rajkumar, 2020), thereby excelling in capturing detailed features at the expense of increased model latency. Lastly, voxel-point based methods (Yang et al., 2019; Shi et al., 2020) aim to merge the computational advantages of voxel-based techniques with the detailed accuracy of point-based methods, marking a progressive trend in this field. Diverging from existing contexts,

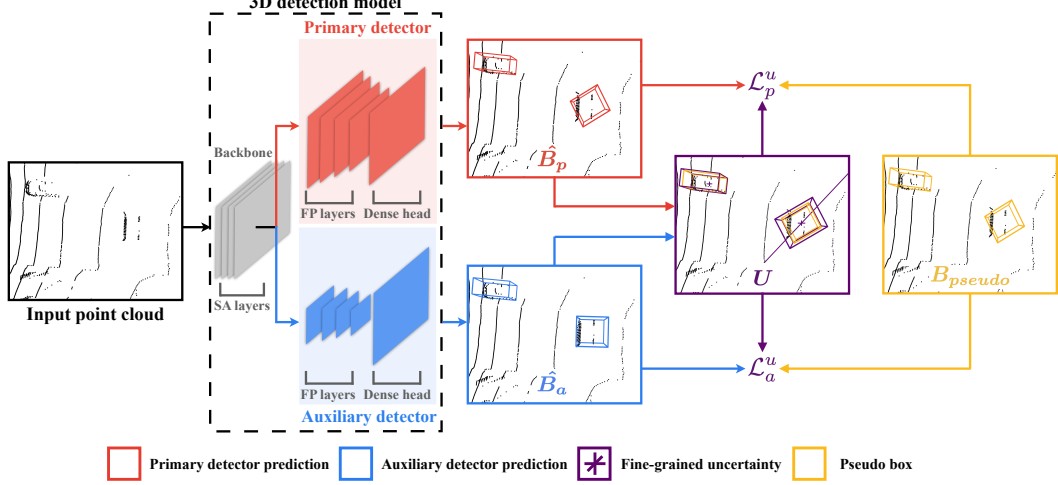

Figure 2: **Overall pipeline.** Given an input point cloud, an auxiliary detector predicts the bounding boxes $\hat{B}_a$ concurrently with the primary detector predictions $\hat{B}_p$. We leverage the discrepancy between the two detector predictions as the uncertainty indicator $U$. Specifically, high coordinate-level uncertainty is assigned to inaccurate pseudo box coordinates. For uncertainty regularization, the original detection loss is rectified by the estimated uncertainty as $\mathcal{L}_p^u$ and $\mathcal{L}_a^u$, reducing the weight of inaccurate pseudo boxes on coordinate level. Note: SA refers to Set Abstraction, and FP refers to Feature Propagation. We insert auxiliary detector after sa_layer_4 in PointRCNN backbone. For *uncertainty visualization*, **purple box** represents the uncertainty of length, width, and height, *i.e.*, $\Delta_l$, $\Delta_w$, and $\Delta_h$; **purple orthogonal lines** indicate the uncertainty of the x, y, and z positions, *i.e.*, $\Delta_x$, $\Delta_y$, and $\Delta_z$; and **purple diagonal line** denotes the uncertainty of orientation, *i.e.*, $\Delta_\theta$. We present a detailed explanation of our uncertainty visualization scheme in Fig. 6. In this example, orientation of pseudo box on the right is inaccurate. Our method assigns high uncertainty for the orientation and reduces its weight during model training.

we attempt to enhance the efficacy of base detection framework (Shi et al., 2019) in an unsupervised setting with fine-grained uncertainty learning.

## 3 METHOD

### 3.1 UNCERTAINTY ESTIMATION

Our approach of uncertainty estimation employs an auxiliary detector architecture (see Fig. 2). Typically, 3D object detection models (Shi et al., 2019; Shi & Rajkumar, 2020) consist of 3D backbone extracting features from point clouds, and 3D detection heads to generate predicted 3D boxes from these features. We introduce an additional 3D detection branch appended to an intermediate layer of the feature extraction backbone. The auxiliary branch mirrors the structure of original branch but differs in channel configuration. We refer to this branch as the auxiliary detector and the original branch is termed the primary detector. We estimate uncertainty as the prediction difference between these two detectors, which can be considered as the degree of disagreement between two different minds. In practice, we use the dense outputs from both detectors, which provide point-wise box predictions across the entire point cloud. For uncertainty estimation, we calculate the $\ell_1$ difference between the point-wise predicted boxes of the primary and auxiliary detectors. This difference is computed at the coordinate level to quantify fine-grained uncertainty:

$$
\begin{aligned}
&\Delta_x = |x_p - x_a|, \Delta_y = |y_p - y_a|, \Delta_z = |z_p - z_a|, \\
&\Delta_l = |l_p - l_a|, \Delta_w = |w_p - w_a|, \Delta_h = |h_p - h_a|, \Delta_\theta = |\theta_p - \theta_a|,
\end{aligned}
\tag{1}
$$

where $x_p, y_p, z_p, l_p, w_p, h_p, \theta_p \in \mathbb{R}^{n \times 1}$ refer to different coordinate vectors of primary detector dense prediction, namely x, y, z for 3D position, length, width, height, and orientation, $x_a, y_a, z_a, l_a, w_a, h_a, \theta_a \in \mathbb{R}^{n \times 1}$ denote coordinate vectors of auxiliary detector dense prediction, $\Delta_x, \Delta_y, \Delta_z, \Delta_l, \Delta_w, \Delta_h, \Delta_\theta \in \mathbb{R}^{n \times 1}$ are estimated uncertainty vectors of different coordinates based on prediction discrepancy between two detectors, and $n$ indicates the number of boxes which is same as the number of points in the point cloud. Furthermore, $\hat{B}_p = [x_p, y_p, z_p, l_p, w_p, h_p, \theta_p] \in \mathbb{R}^{n \times 7}$ refers to primary detector dense predictions, $\hat{B}_a =$

$[\boldsymbol{x_a}, \boldsymbol{y_a}, \boldsymbol{z_a}, \boldsymbol{l_a}, \boldsymbol{w_a}, \boldsymbol{h_a}, \boldsymbol{\theta_a}] \in \mathbb{R}^{n \times 7}$ denotes auxiliary detector dense predictions, and $\boldsymbol{U} = [\boldsymbol{\Delta_x}, \boldsymbol{\Delta_y}, \boldsymbol{\Delta_z}, \boldsymbol{\Delta_l}, \boldsymbol{\Delta_w}, \boldsymbol{\Delta_h}, \boldsymbol{\Delta_\theta}] \in \mathbb{R}^{n \times 7}$ represents the estimated fine-grained uncertainty. Notably, each coordinate of the 3D box is assigned an estimated value, which reflects the uncertainty of that specific coordinate.

**Discussions. Why can uncertainty estimation reflect the inaccuracy of pseudo boxes?** Accurate pseudo boxes are well-aligned with the object regions in the input point cloud, typically exhibiting consistent characteristics such as tightly enclosing specific point groups and maintaining a reasonable size. In contrast, inaccurate pseudo boxes show significant and unpredictable variations, making them harder to interpret. This inherent uncertainty can confuse the model, leading to highly varying predictions for the same object. Consequently, discrepancies between the two detector predictions indicate elevated uncertainty, reflecting the inaccuracy of pseudo boxes. **Why choose dense predictions for uncertainty estimation instead of using predictions from the Region-of-Interest (ROI) head?** Since the dense outputs predict a box for each point in the point cloud, they generate the same number of predictions regardless of the model structure, ensuring consistency between primary and auxiliary detectors. This consistency naturally simplifies the calculation of differences between two detector predictions for estimate uncertainty. In 3D detection model (Shi et al., 2019), ROI head aggregates point-wise predictions into certain numbers of final bounding boxes, and the numbers of predicted boxes can vary between the primary and auxiliary detectors. While it is feasible to utilize the output from ROI head for uncertainty estimation, the different numbers of boxes from primary and auxiliary detectors require a matching process. Matching boxes between two detectors introduces significant computational overhead. Given the additional training cost, we choose not to rely on the predictions from ROI head. **Why use an auxiliary detector to estimate uncertainty, instead of directly regressing uncertainty, as done in previous works (Choi et al., 2019; He et al., 2019)?** We have studied the additional channel method, which involves using extra channels to regress the uncertainty. However, this approach did not yield satisfactory results, as it suffers from overfitting issues, such as predicting zero uncertainty for all samples or uniformly high uncertainty. We attribute this to the inherent complexity of unsupervised 3D detection: simply adding extra channels introduces too few model parameters to effectively capture uncertainty, which is insufficient to manage the complexities involved.

### 3.2 Uncertainty Regularization

Upon deriving the fine-grained uncertainty, we employ it to refine the iterative learning process. Our objective is to adaptively reduce the negative effects of inaccurate pseudo boxes at coordinate level. To achieve this, we rectify original detection loss by incorporating our estimated uncertainty:

$$\mathcal{L}_p^u = \sum_{i=1}^{7} (\frac{\mathcal{L}_{p,i}}{\exp{(\boldsymbol{U_i})}} + \lambda \cdot \boldsymbol{U_i}), \quad \mathcal{L}_a^u = \sum_{i=1}^{7} (\frac{\mathcal{L}_{a,i}}{\exp{(\boldsymbol{U_i})}} + \lambda \cdot \boldsymbol{U_i}), \tag{2}$$

where $\mathcal{L}_p^u, \mathcal{L}_a^u$ denote the uncertainty-regularized loss of primary and auxiliary detectors. For brevity, we represent 7 coordinates of 3D box (see Eq. 1) by $i = 1, 2, ..., 7$. $\mathcal{L}_{p,i}, \mathcal{L}_{a,i}$ represent the original dense head losses of primary and auxiliary detectors for the $i$-th coordinate, which are calculated by the $\ell_1$ loss between corresponding coordinate of the predicted boxes and pseudo boxes. Specifically, $\mathcal{L}_{p,i} = |\hat{\boldsymbol{B}}_{\boldsymbol{p,i}} - \boldsymbol{B_{pseudo,i}}|, \mathcal{L}_{a,i} = |\hat{\boldsymbol{B}}_{\boldsymbol{a,i}} - \boldsymbol{B_{pseudo,i}}|$, where $\boldsymbol{B_{pseudo,i}} \in \mathbb{R}^{n \times 1}$ is the $i$-th coordinate of assigned dense pseudo boxes. $\boldsymbol{U_i}$ denotes the estimated fine-grained uncertainty of the corresponding coordinate in $\boldsymbol{U}$. To prevent divide-by-zero errors and stabilize the learning process, we normalize estimated uncertainty with exponential function. Additionally, we incorporate term $\lambda \cdot \boldsymbol{U_i}$ to prevent the model from consisting predicting high uncertainty, where $\lambda$ controls penalty strength. Empirically, when uncertainty of certain coordinate is high, weight of that inaccurate pseudo box coordinate is diminished, thereby reducing its impact on training process. Conversely, when uncertainty is low, for instance, nearing zero, the loss reverts to original detection loss, preserving the full influence of that pseudo box coordinate. As a result, our uncertainty regularization dynamically mitigates negative effects of inaccurate pseudo boxes on coordinate level.

The regularization process is uniformly applied to both primary and auxiliary detectors. Each detector takes into account the prediction of the other and adjusts weights of pseudo box coordinates accordingly, who diminishes influence of pseudo box coordinates when significant prediction disagreement is evident, and reserves impact of pseudo box coordinates when two predictions concur.

Therefore, the final loss $\mathcal{L}_{total}$ can be formulated as:

$$\mathcal{L}_{total} = \mathcal{L}_p^u + \mu \cdot \mathcal{L}_a^u, \tag{3}$$

where $\mathcal{L}_p^u$ is the uncertainty-regularized loss for the primary detector, $\mathcal{L}_a^u$ is the uncertainty-regularized loss for the auxiliary detector, $\mu$ denotes the auxiliary detector loss weight.

**Discussions. Why is uncertainty regularization fine-grained?** Our calculation process operates at the box coordinate level. This allows our method to identify coordinate-specific inaccuracies in pseudo boxes and dynamically mitigate their negative influence. During the pseudo box generation process, pseudo boxes can exhibit inaccuracies in specific coordinates, such as only in the orientation angle. In such cases, treating the entire box as fully certain or uncertain is not reasonable. Our fine-grained regularization approach can selectively reduce the negative influence of the inaccurate coordinate while preserving the efficacy of other accurate coordinates. **Why not use rule-based uncertainty?** Our uncertainty-aware framework is learnable and more adaptive. There are methods (Wu et al., 2024) where uncertainty in pseudo boxes is determined using fixed rules based on factors like distance, the number of points in the box, or the distribution pattern of points within the box. These rules are devised based on human-observed knowledge, *e.g.*, the further the box, the higher the uncertainty. However, such rules can lead to errors. For example, a distant box can be very accurate, but under rule-based uncertainty, its influence can be unjustly diminished, potentially degrading model performance. Our learnable uncertainty avoids this pitfall by not only assimilating human-observed rules and knowledge but also adaptively handling different cases. For instance, if a distant pseudo box is very accurate, both the primary and auxiliary detectors can provide similar predictions, resulting in low uncertainty and ensuring that the box is appropriately valued during training. **What differentiates our work from the model ensemble approaches (Sagi & Rokach, 2018)?** We focus on improving the performance of a single model. Our final detection results benefit from regularization gained from both the primary and auxiliary detectors. During the inference phase, we only enable the primary detector, rather than typical model ensemble approaches that aggregate multiple different models. Notably, our approach is also scalable and can be applied to individual models within an ensemble, if desired.

## 4 EXPERIMENT

### 4.1 SETTINGS

**Datasets.** Our experiments are conducted using the nuScenes (Caesar et al., 2020) and Lyft (Houston et al., 2021) datasets, adhering to the settings established by MODEST (You et al., 2022). We consider data samples that meet the multi-traversal requirements, *i.e.*, point clouds collected at locations traversed more than once by the data-collecting vehicle. On nuScenes, we obtain 3,985 point clouds for training and 2,412 for testing. Similarly, we utilize 11,873 training and 4,901 testing point clouds on Lyft. It is worth noting that we do not use any ground truth 3D boxes during the training phase and ground truth boxes are exclusively used for evaluation.

**Backbone.** The primary backbone for our 3D detection is PointRCNN (Shi et al., 2019). PointR-CNN utilizes PointNet++ (Qi et al., 2017) for extracting point-wise features from the LiDAR point clouds. Within PointNet++, Set Abstraction layers first perform point grouping and local feature extraction, Feature Propagation layers then conduct feature upsampling and propagate abstract features back to point-wise representation. Following this, dense head predicts a 3D box for each point based on these extracted features. Lastly, region of interest (ROI) head aggregates object proposals from the point-wise predictions into final predictions.

**Implementation Details.** For construction of auxiliary detector, we first incorporate 4 additional Feature Propagation layers after the last Set Abstraction layer in PointRCNN. These layers mirror the structure of the original Feature Propagation layers but with varied channel numbers. Specifically, the channel numbers in the original Feature Propagation layers are $(C_1, C_2, C_3, C_4)$, while in the introduced Feature Propagation layers, they are scaled to $(\gamma \cdot C_1, \gamma \cdot C_2, \gamma \cdot C_3, \gamma \cdot C_4)$, where $\gamma$ represents coefficient to adjust the channel number in the introduced Feature Propagation layers. In practice, the adopted $(C_1, C_2, C_3, C_4)$ are (128, 256, 512, 512) and $\gamma = 0.5$ yields the best results. We then integrate a new dense head and ROI head after the introduced Feature Propagation layers to establish the auxiliary detector. We follow the self training paradigm established by previous work MODEST (You et al., 2022). Specifically, we conduct seed training and 10 rounds of self training

Table 1: **Quantitative results on nuScenes (Caesar et al., 2020) and Lyft (Houston et al., 2021).** We report $AP_{BEV}$ and $AP_{3D}$ at $IoU = 0.25$ for objects across various distances, presented in the format $AP_{BEV}$ / $AP_{3D}$. $T = 0$ indicates training from seed boxes, while $T = 2$ and $T = 10$ correspond to the results from the $2th$ and $10th$ round of self-training, respectively. Supervised performance of model trained with ground-truth boxes is in the first row (Supervised). $^*$ denotes our reproduced results by adhering to official settings, which include two rounds of self-training. (a) Detection results on nuScenes. It is worth noting that our UA3D significantly surpasses the state-of-the-art OYSTER (Zhang et al., 2023) across all evaluated metrics. This validates the efficacy of our proposed coordinate-level uncertainty estimation and regularization in mitigating negative impacts of noisy pseudo boxes for unsupervised 3D object detection. (b) Detection results on Lyft. Our UA3D significantly outperforms MODEST (You et al., 2022) by 4.1% in $AP_{BEV}$ and 2.0% in $AP_{3D}$. Notably, we employ same hyper-parameters as those used in nuScenes experiments and observe a consistent improvement.

(a)

| Method | T | 0-30m | 30-50m | 50-80m | 0-80m |
|---|---|---|---|---|---|
| Supervised | - | 39.8 / 34.5 | 12.9 / 10.0 | 4.4 / 2.9 | 22.2 / 18.2 |
| MODEST-PP | 0 | 0.7 / 0.1 | 0.0 / 0.0 | 0.0 / 0.0 | 0.2 / 0.1 |
| MODEST-PP | 2 | - | - | - | - |
| MODEST | 0 | 16.5 / 12.5 | 1.3 / 0.8 | 0.3 / 0.1 | 7.0 / 5.0 |
| MODEST | 10 | 24.8 / 17.1 | 5.5 / 1.4 | 1.5 / 0.3 | 11.8 / 6.6 |
| OYSTER | 0 | 14.7 / 12.3 | 1.5 / 1.1 | 0.5 / 0.3 | 6.2 / 5.4 |
| OYSTER | 2* | 26.6 / 19.3 | 4.4 / 1.8 | 1.7 / 0.4 | 12.7 / 8.0 |
| UA3D (ours) | 0 | 13.7 / 11.5 | 0.9 / 0.6 | 0.5 / 0.2 | 5.4 / 4.9 |
| UA3D (ours) | 10 | **38.3 / 23.8** | **10.1 / 3.5** | **4.3 / 0.7** | **19.6 / 10.5** |

(b)

| Method | T | 0-30m | 30-50m | 50-80m | 0-80m |
|---|---|---|---|---|---|
| Supervised | - | 82.8 / 82.6 | 70.8 / 70.3 | 50.2 / 49.6 | 69.5 / 69.1 |
| MODEST-PP | 0 | 46.4 / 45.4 | 16.5 / 10.8 | 0.9 / 0.4 | 21.8 / 18.0 |
| MODEST-PP | 10 | 49.9 / 49.3 | 32.3 / 27.0 | 3.5 / 1.4 | 30.9 / 27.3 |
| MODEST | 0 | 65.7 / 63.0 | 41.4 / 36.0 | 8.9 / 5.7 | 42.5 / 37.9 |
| MODEST | 10 | 73.8 / **71.3** | 62.8 / 60.3 | 27.0 / 24.8 | 57.3 / 55.1 |
| UA3D (ours) | 0 | 66.0 / 63.3 | 43.8 / 36.3 | 8.9 / 5.1 | 43.2 / 38.0 |
| UA3D (ours) | 10 | **74.1** / 71.2 | **63.6 / 61.7** | **36.8 / 29.0** | **61.4 / 57.1** |

in all our experiments. In seed training, initially generated pseudo boxes from clustering algorithms are used to bootstrap a detection model. Afterward, in each self training round, trained model from previous round is first utilized to infer on training set to generate pseudo boxes, and new model is trained based on those model-inferred boxes. For both nuScenes and Lyft, the regularization coefficient $\lambda$ is set to $1e^{-5}$. We train 80 epochs for nuScenes and 60 epochs for Lyft. We use Adam as the optimizer with a learning rate of 0.01, weight decay of 0.01, and momentum of 0.9. The learning rate is decayed at epochs 35 and 45 with a decay rate of 0.1. The batch size is set to 2 per GPU. We apply gradient norm clipping of 10. Following the settings of previous work (You et al., 2022), we sample 6,144 points per point cloud in nuScenes and 12,288 points per point cloud in Lyft to enhance computational efficiency. We utilize 4 A6000 (48G) GPUs for all our experiments.

## 4.2 COMPARISON WITH STATE-OF-THE-ART METHODS

We present the results for nuScenes (Caesar et al., 2020) in Table 1a. Our uncertainty-aware framework outperforms the state-of-the-art method OYSTER (Zhang et al., 2023) by 6.9% in $AP_{BEV}$ and 2.5% in $AP_{3D}$, respectively. This performance enhancement underscores the efficacy of our proposed uncertainty-aware method in refining learning process from noisy pseudo boxes. It confirms that reducing the negative impact of inaccurate pseudo boxes on coordinate level can significantly boost model detection performance. Notably, for objects in the long-range (50-80m), $AP_{BEV}$ sees a remarkable increase of 253% (from 1.7% to 4.3%). This significant boost is attributed to the typically lower accuracy of long-range pseudo boxes, where uncertainty plays a pivotal role in dynamically adjusting the weights of pseudo boxes coordinates according to their varying qualities.

We further conduct experiments on Lyft (Houston et al., 2021) (see Table 1b). Our uncertainty-aware method surpasses MODEST by 4.1% in $AP_{BEV}$ and 2.0% in $AP_{3D}$. Notably, we use the same hyper-parameter settings as those in nuScenes experiments, validating the generalizability and effectiveness of our uncertainty-aware approach. The most significant improvements are also observed in the long-range (50-80m), with increases of 9.8% in $AP_{BEV}$ and 4.2% in $AP_{3D}$. This verifies the efficacy of our method in enhancing the detection capability of distant objects, which are typically challenging to recognize.

## 4.3 ABLATION STUDIES AND FURTHER DISCUSSION

**Comparison with Rule-Based Uncertainty.** We compare our proposed learnable uncertainty-aware method with rule-based uncertainty to validate the superiority of our learnable approach (see Table 2a). We implement several rule-based uncertainties as our baselines, encompassing distance-based, number-of-points-in-box-based (Numpts-based), and volume-based uncertainty. We follow

Table 2: **Ablation studies on the nuScenes dataset.** We report $AP_{BEV}$ and $AP_{3D}$ at $IoU = 0.25$ for objects across various distances. (a) Ablation study of rule-based uncertainty and our proposed learnable uncertainty-aware framework. Our learnable uncertainty surpasses all types of rule-based uncertainty, validating its superiority in handling complex cases where rule-based uncertainty can fail. (b) Ablation study of the uncertainty granularity. We find that our proposed coordinate-level uncertainty outperforms other coarse-grained uncertainty, such as box-level and point cloud-level. By addressing the inaccuracies in box coordinates individually, our coordinate-level uncertainty reduces the negative impact of noisy pseudo boxes more adaptively. (c) Ablation study on the auxiliary detector structure. $\gamma$ denotes the channel number coefficient of the auxiliary detector, with the best performance achieved at 0.5. Being slightly smaller than the primary detector, auxiliary detector can accurately fit correct pseudo boxes while avoiding over-fitting to noisy ones. This setting enhances the identification of inaccurate pseudo boxes, effectively unlocking the potential of our uncertainty-aware framework. (d) Ablation study on $\lambda$. We obtain the best result at $\lambda = 1e^{-5}$ as it ensures uncertainty estimation and regularization play a proper role, preventing the uncertainty from vanishing or exploding.

(a)

| Method | 0-30m | 30-50m | 50-80m | 0-80m |
|---|---|---|---|---|
| Distance-based | 29.6 / 19.6 | 7.2 / 2.2 | 3.2 / 0.5 | 14.8 / 8.1 |
| Numpts-based | 27.3 / 17.6 | 7.3 / 2.8 | 2.3 / 0.3 | 13.7 / 7.5 |
| Volume-based | 25.7 / 17.7 | 5.6 / 2.2 | 2.5 / 0.4 | 12.3 / 7.4 |
| UA3D (ours) | **38.3 / 23.8** | **10.1 / 3.5** | **4.3 / 0.7** | **19.6 / 10.5** |

(b)

| Granuity | 0-30m | 30-50m | 50-80m | 0-80m |
|---|---|---|---|---|
| Coordinate-level | 38.3 / 23.8 | 10.1 / 3.5 | 4.3 / 0.7 | **19.6 / 10.5** |
| Box-level | 34.9 / 24.6 | 7.5 / 2.8 | 3.6 / 0.1 | 17.2 / 9.9 |
| Point cloud-level | 27.7 / 18.7 | 3.6 / 1.2 | 1.2 / 0.1 | 12.1 / 6.7 |

(c)

| $\gamma$ | 0-30m | 30-50m | 50-80m | 0-80m |
|---|---|---|---|---|
| 0.25 | 32.6 / 23.5 | 8.6 / 3.1 | 4.3 / 0.2 | 16.9 / 9.9 |
| 0.5 | 38.3 / 23.8 | 10.1 / 3.5 | 4.3 / 0.7 | **19.6 / 10.5** |
| 1 | 29.6 / 22.3 | 6.0 / 2.3 | 3.3 / 0.1 | 14.7 / 8.5 |
| 2 | 29.5 / 20.5 | 7.9 / 3.0 | 4.4 / 0.3 | 15.8 / 8.9 |

(d)

| $\lambda$ | 0-30m | 30-50m | 50-80m | 0-80m |
|---|---|---|---|---|
| $1e^{-4}$ | 33.8 / 20.4 | 6.1 / 1.5 | 2.9 / 0.3 | 15.2 / 7.4 |
| $1e^{-5}$ | 38.3 / 23.8 | 10.1 / 3.5 | 4.3 / 0.7 | **19.6 / 10.5** |
| $1e^{-6}$ | 18.1 / 13.7 | 3.2 / 1.3 | 1.6 / 0.2 | 8.4 / 5.6 |

common human observed rules, *e.g.*, the farther the pseudo box is, the fewer points the pseudo box contains, or the smaller the pseudo box is, the less accurate and more uncertain it becomes. For distance-based uncertainty, the uncertainty of a pseudo box is quantified as $u = \frac{\min(b_x, \tau_x)}{\tau_x}$, where $b_x$ denotes the distance of the box from the ego vehicle, and $\tau_d$ represents the selected distance threshold. We assign a constant uncertainty value of 1 for boxes located beyond $\tau_x$, which we empirically set at $\tau_x = 100m$. For numpts-based uncertainty, the uncertainty is formulated as $u = \frac{\tau_n}{\min(b_{num\_pts}, \tau_n)}$, where $b_{num\_pts}$ refers to the number of points within the 3D pseudo box, and $\tau_n$ is the selected points threshold set at $\tau_n = 100$. For volume-based uncertainty, the uncertainty is computed as $u = \frac{\tau_v}{\min(b_l \cdot b_w \cdot b_h, \tau_v)}$, where $b_l$, $b_w$, and $b_h$ indicate the length, width, and height of the 3D pseudo box, and $\tau_v$ is the chosen volume threshold set at $\tau_v = 10m^3$. The uncertainty for each pseudo box is calculated during training and utilized to regularize the original detection loss. Our learnable uncertainty consistently outperforms all rule-based uncertainties by effectively addressing scenarios where rule-based approaches fail. For instance, a box with a high number of points is typically assumed to have low uncertainty, but can be inaccurate. Our learnable uncertainty is capable of assigning high uncertainty to such cases due to prediction discrepancies between the primary and auxiliary detectors.

**Ablation of Different Granularities.** We present an ablation study on the uncertainty granularity in Table 2b. For our proposed coordinate-level uncertainty, the uncertainty estimation and regularization is applied at the coordinate level, where the loss weight for each coordinate of each box is adjusted adaptively based on its uncertainty value. For box-level uncertainty, we sum the uncertainty values of the 7 coordinates for each box, using this sum as the overall uncertainty for the box. Concurrently, the loss values of all 7 coordinates are combined into a total loss for the box, and this total loss is regularized with the corresponding box uncertainty. For point cloud-level uncertainty, we aggregate the uncertainty of all boxes in the point cloud to represent the overall uncertainty of the point cloud. Meanwhile, the losses of all boxes in the point cloud are summed into an overall loss, which is then regularized by the corresponding point cloud-level uncertainty. We observe that the best results are achieved with our coordinate-level uncertainty. This approach corrects inaccurate pseudo boxes in a more fine-grained and adaptive manner, effectively mitigating the negative impact of noise. In contrast, box-level uncertainty regularization treats the entire box as either certain or uncertain, ignoring differences among the coordinates. For example, a box can have an inaccurate

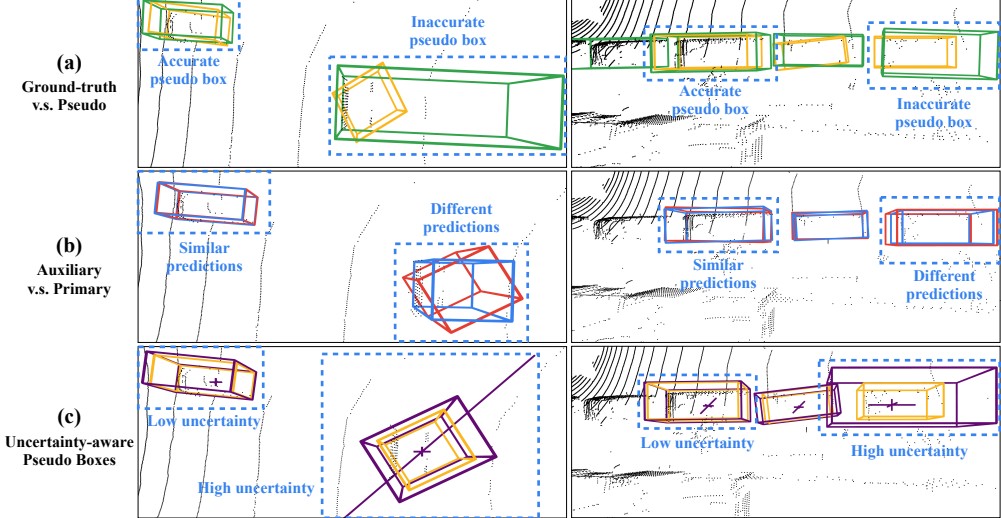

Figure 3: **Correspondence between pseudo label inaccuracy and high uncertainty.** (a) We present ground truth and pseudo boxes in two different point clouds (left and right columns). Each point cloud contains both accurate and inaccurate pseudo boxes. We observe that pseudo boxes can be significantly inaccurate in terms of the shape, location, and rotation. Direct usage of these boxes for training can easily impair the performance of the detection model. (b) We present the predictions from the primary and auxiliary detectors. Two detector predictions align closely for objects with accurate pseudo boxes but diverge for those with inaccurate ones. The mismatch between inaccurate pseudo boxes and the actual point cloud distribution can confuse the model, resulting in varying interpretations. (c) We present our uncertainty-aware pseudo boxes. Fine-grained coordinate-level uncertainty is estimated, *e.g.*, the orientation uncertainty for the right object (in left column) is high (as indicated by the long **purple diagonal line**), due to its inaccuracy in the pseudo box. The *colors* follow the same conventions in Fig. 2. A detail explanation of our *uncertainty visualization* scheme is shown in Fig. 6.

length while other dimensions are accurate. The coarse-grained box-level approach can compromise the efficacy of regularization. At the point cloud level, the regularization effect is weak, resulting in performance degradation to the baseline (MODEST).

**Design of Uncertainty Estimation.** We present an ablation study on the design of the auxiliary detector in Table 2c. The configuration with $\gamma = 0.5$ yields the best results. This configuration provides enough model capacity to fit accurate pseudo boxes while avoiding over-fitting to noisy pseudo boxes. As a result, the primary and auxiliary detector predictions tend to diverge for inaccurate pseudo boxes, leading to more effective uncertainty estimation and regularization. $\gamma = 0.25$ indicates a smaller auxiliary detector with weaker capacity in fitting pseudo boxes. Other than inaccurate boxes, such a model will also result in higher prediction discrepancies for those accurate boxes and thus impair the uncertainty estimation process. Conversely, larger auxiliary detectors, such as those with $\gamma = 1$ and $\gamma = 2$, exhibit learning capacities similar to the primary detector, which diminishes the efficacy of uncertainty learning.

**Design of Uncertainty Regularization.** We explore the effects of varying the uncertainty regularization coefficient $\lambda$ (see Eq. 2) in Table 2d. The optimal performance is observed with $\lambda = 1e^{-5}$, which allows uncertainty estimation and regularization to play a proper role and avoids uncertainty vanishing or explosion. Other settings yield sub-optimal results compared with $\lambda = 1e^{-5}$. A high $\lambda = 1e^{-4}$ imposes a strong penalty for high uncertainty and suppresses the role of uncertainty during training. Conversely, a low $\lambda = 1e^{-6}$, which imposes a minimal penalty for high uncertainty, leads to excessively high uncertainty values across all samples. This reduces the influence of the original detection loss, resulting in slow learning process.

## 4.4 QUALITATIVE ANALYSIS

We visualize the obtained uncertainty in Fig. 3 and such analysis further validates the correspondence between the pseudo boxes inaccuracies and estimated uncertainty. Specifically, we observe that accurate pseudo boxes, which typically lead to consistent predictions from both the primary and

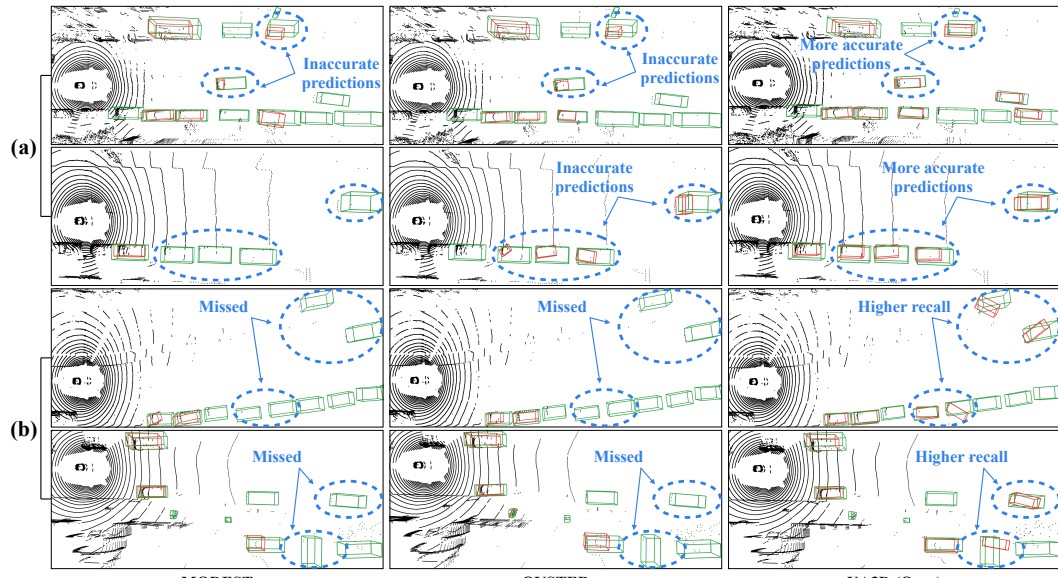

Figure 4: **Visualization comparison between different methods.** We compare the predictions of MODEST (You et al., 2022), OYSTER (Zhang et al., 2023), and our uncertainty-aware framework. **Green** boxes denote ground truth boxes and **red** boxes are predictions. (a) Generally, our method shows a clear improvement in box coordinate accuracy over previous methods. (b) For some challenging objects with few points or far away, our method can still retain a higher recall rate.

auxiliary detectors, exhibit low uncertainty. In contrast, when a pseudo box shows inaccuracies in certain coordinates, the estimated uncertainty for those coordinates is significantly higher since the predictions from the primary and auxiliary detectors diverge on those coordinates.

In Fig 4, we compare the predictions from our uncertainty-aware method against those from MODEST (You et al., 2022) and OYSTER (Zhang et al., 2023). Notably, our method achieves more accurate predictions in terms of shape, location, and orientation (see (a) in Fig.4). This enhancement stems from our learnable uncertainty which reduces the impact of imprecise pseudo boxes at a fine-grained coordinate level. By integrating uncertainty estimation and regularization processes that focus on individual coordinates, our model avoids overfitting to erroneous box coordinates. Furthermore, we observe an increase in the recall rate, especially for distant and smaller objects (see (b) in Fig.4). The pseudo boxes for these objects are often less reliable due to the challenges in estimating such boxes. Our approach selectively discounts these unreliable boxes, allowing high-quality boxes to play a more prominent role. Consequently, our model benefits more from accurate pseudo boxes of challenging objects, enhancing recall performance for these categories.

## 5 CONCLUSION

In this paper, we aim to mitigate the negative impact of inaccurate pseudo boxes in unsupervised 3D object detection. Direct usage of those inaccurate pseudo boxes can significantly impair model performance. To address this issue, we propose an uncertainty-aware framework that identifies the inaccuracy of pseudo boxes at a fine-grained coordinate level and reduces their negative effect. In uncertainty estimation phase, we introduce an auxiliary detector to capture the prediction discrepancy with the primary detector, harnessing these discrepancies as fine-grained indicators of uncertainty. In uncertainty regularization phase, the estimated uncertainty is utilized to refine the training process, adaptively minimizing the negative effects of inaccurate pseudo boxes at the coordinate level. Quantitative experiments on nuScenes and Lyft validate the effectiveness of our uncertainty-aware framework. Additionally, qualitative results show the superiority of our method and reveal the correlation between high uncertainty and pseudo label inaccuracy.

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

# A APPENDIX

## A.1 IMPLEMENTATION DETAILS

**Hyper-parameters.** For nuScenes (Caesar et al., 2020), the batch size is set to 2 per GPU. Training is conducted for 80 epochs using the Adam optimizer with a one-cycle policy. The initial learning rate is 0.01, with a weight decay of 0.01 and a momentum of 0.9. Learning rate decay is applied at epochs 35 and 45 with a decay rate of 0.1. Additionally, a learning rate clip of $1e^{-7}$ and a gradient norm clip of 10 are employed. We perform one round of seed training followed by 10 rounds of self-training for all experiments. Each round of training takes approximately 4 hours, resulting in a total training time of about 44 hours (4 hours $\times$ 11 rounds). For Lyft (Houston et al., 2021), we reduce the number of epochs to 60 for efficiency, considering that the Lyft dataset is 3 times larger than nuScenes (You et al., 2022). The self-training pipeline for Lyft also consists of one round of seed training and 10 rounds of self-training. Each training round takes approximately 12 hours, leading to a total training time of around 131 hours (12 hours $\times$ 11 rounds). Other settings remain the same as those for nuScenes, without specific tuning, to validate the generalizability of our proposed uncertainty-aware framework.

**Data Processing.** For both nuScenes and Lyft, we apply several data augmentations. We sample 6,144 points per point cloud for nuScenes, while for Lyft, we sample 12,288 points per point cloud, as the point clouds in Lyft are generally denser than those in nuScenes. We perform random world flipping of the entire point cloud along the x-axis. We also apply random world rotation within the angle range of [-0.785, 0.785] and random world scaling within the scale ratio range of [0.95, 1.05]. Point shuffling is applied to the training set but not to the test set. We focus on object discovery, following the trajectory of previous works such as MODEST, OYSTER, and LiSe. We do not explicitly consider object categories during the experiments.

**Self-training Pipeline.** Our uncertainty-aware framework operates within a self-training pipeline, mainly based on the settings outlined in MODEST. In general, a self-training pipeline consists of two stages: seed training and self-training. Initial generated pseudo boxes are referred to as seeds. During seed training, an initial detection model is trained based on those seeds. Different from seed training, in self-training, trained model from previous round is first applied to the training set to obtain refined pseudo boxes. Following this, a new detection model is trained on the refined pseudo boxes. The process is iteratively repeated for $T$ rounds.

## A.2 MODEL STRUCTURE

**Overall Model Structure.** The detection model we use is PointRCNN, which utilizes PointNet++ for point-wise feature extraction. After feature extraction, the dense head predicts a box for each point. Following this, the ROI head aggregates these point-wise predictions and applies score thresholds to produce the final predictions. PointNet++ mainly comprises Set Abstraction Layers and Feature Propagation Layers. The Set Abstraction Layers group the entire point cloud into local regions, where local features are extracted using PointNet to capture geometric structures. By stacking multiple Set Abstraction Layers with varying neighborhood sizes, a hierarchical representation of the point cloud is built, allowing the model to learn more fine-grained and complex features at multiple scales. Based on this hierarchical representation, the Feature Propagation Layers iteratively upsample and propagate features back to the original point-wise level, recovering detailed information to support various downstream tasks. For the introduced auxiliary detection branch, we introduce additional Feature Propagation Layers into the middle of the PointNet++ feature extraction backbone. These layers are attached to the final layer of the original Set Abstraction Layers and have a similar structure but differ in the number of channels. New dense head and ROI head are also introduced to generate auxiliary detector predictions based on the features extracted from the added Feature Propagation Layers. These added dense head and ROI head are designed with different input channels to accommodate the modified channel dimensions of the newly added Feature Propagation Layers.

**Detailed Model Settings.** We present a detailed description of our model structure in Table 3. The shared feature extraction backbone consists of 4 SA layers. The primary detection branch follows the original PointRCNN model, while the auxiliary detection branch is newly added. This auxiliary branch is attached to the last SA layer of the shared backbone, with its channel numbers halved compared to the primary detection branch. The prediction discrepancy between the primary and

Table 3: **Detailed model structure.** The SALayer refers to the Set Abstraction Layer, which performs point grouping and local feature extraction. The Grouper is a rule-based operation for point cloud grouping, typically based on Farthest Point Sampling (FPS). The ConvBlock is a Convolutional Block composed of a convolutional layer, a batch normalization layer, and a ReLU layer. The FPLayer refers to the Feature Propagation layer, which performs feature upsampling and propagates abstract features back to each point in the point cloud. The DenseHead predicts one box for each point in the cloud. The LinearBlock consists of a linear layer, a batch normalization layer, and a ReLU layer. The WeightedSmoothL1Loss is an updated version of the L1 loss that applies different weights to different coordinates.

| Shared Feature Extraction Backbone | |
|:---:|:---:|
| SALayer1: | |
| Grouper | |
| ConvBlock(4, 16) , ConvBlock(16, 16) , ConvBlock(16, 32) | |
| ConvBlock(4, 32) , ConvBlock(32, 32) , ConvBlock(32, 64) | |
| SALayer2: | |
| Grouper | |
| ConvBlock(99, 64) , ConvBlock(64, 64) , ConvBlock(64, 128) | |
| ConvBlock(99, 64) , ConvBlock(64, 96) , ConvBlock(96, 128) | |
| SALayer3: | |
| Grouper | |
| ConvBlock(259, 128) , ConvBlock(128, 196) , ConvBlock(196, 256) | |
| ConvBlock(259, 128) , ConvBlock(128, 196) , ConvBlock(196, 256) | |
| SALayer4: | |
| Grouper | |
| ConvBlock(515, 256) , ConvBlock(256, 256) , ConvBlock(256, 512) | |
| ConvBlock(515, 256) , ConvBlock(256, 384) , ConvBlock(384, 512) | |
| **Primary Detection Branch** | **Auxiliary Detection Branch** |
| FPLayer1: | FPLayer1: |
| ConvBlock(257, 128) , ConvBlock(128, 128) | ConvBlock(129, 64) , ConvBlock(64, 64) |
| FPLayer2: | FPLayer2: |
| ConvBlock(608, 256) , ConvBlock(256, 256) | ConvBlock(352, 128) , ConvBlock(128, 128) |
| FPLayer3: | FPLayer3: |
| ConvBlock(768, 512) , ConvBlock(512, 512) | ConvBlock(512, 256) , ConvBlock(256, 256) |
| FPLayer4: | FPLayer4: |
| ConvBlock(1536, 512) , ConvBlock(512, 512) | ConvBlock(1536, 256) , ConvBlock(256, 256) |
| DenseHead: | DenseHead: |
| LinearBlock(128, 256) | LinearBlock(64, 256) |
| LinearBlock(256, 256) | LinearBlock(256, 256) |
| LinearBlock(256, 8) | LinearBlock(256, 8) |
| WeightedSmoothL1Loss | WeightedSmoothL1Loss |
| ROIHead: | ROIHead: |
| ProposeLayer | ProposeLayer |
| SALayer1((131, 128), (128, 128), (128, 128)) | SALayer1((67, 128), (128, 128), (128, 128)) |
| SALayer2((131, 128), (128, 128), (128, 256)) | SALayer2((131, 128), (128, 128), (128, 256)) |
| SALayer3((259, 256), (256, 256), (256, 512)) | SALayer3((259, 256), (256, 256), (256, 512)) |
| XYZUPLayer | XYZUPLayer |
| ConvBlock(5, 128) , ConvBlock(128, 128) | ConvBlock(5, 64) , ConvBlock(64, 64) |
| MergeDownLayer | MergeDownLayer |
| ConvBlock(256, 128) | ConvBlock(128, 64) |
| RegressionLayer((512, 256), (256, 256), (256, 7)) | RegressionLayer((512, 256), (256, 256), (256, 7)) |
| WeightedSmoothL1Loss | WeightedSmoothL1Loss |

auxiliary detectors allows us to identify uncertainty in noisy pseudo boxes during unsupervised 3D object detection.

## A.3 More qualitative results

We present additional qualitative results in Fig. 5. As shown in Fig. 5 (a), our uncertainty-aware framework generates more accurate predictions regarding object shape, location, and orientation.

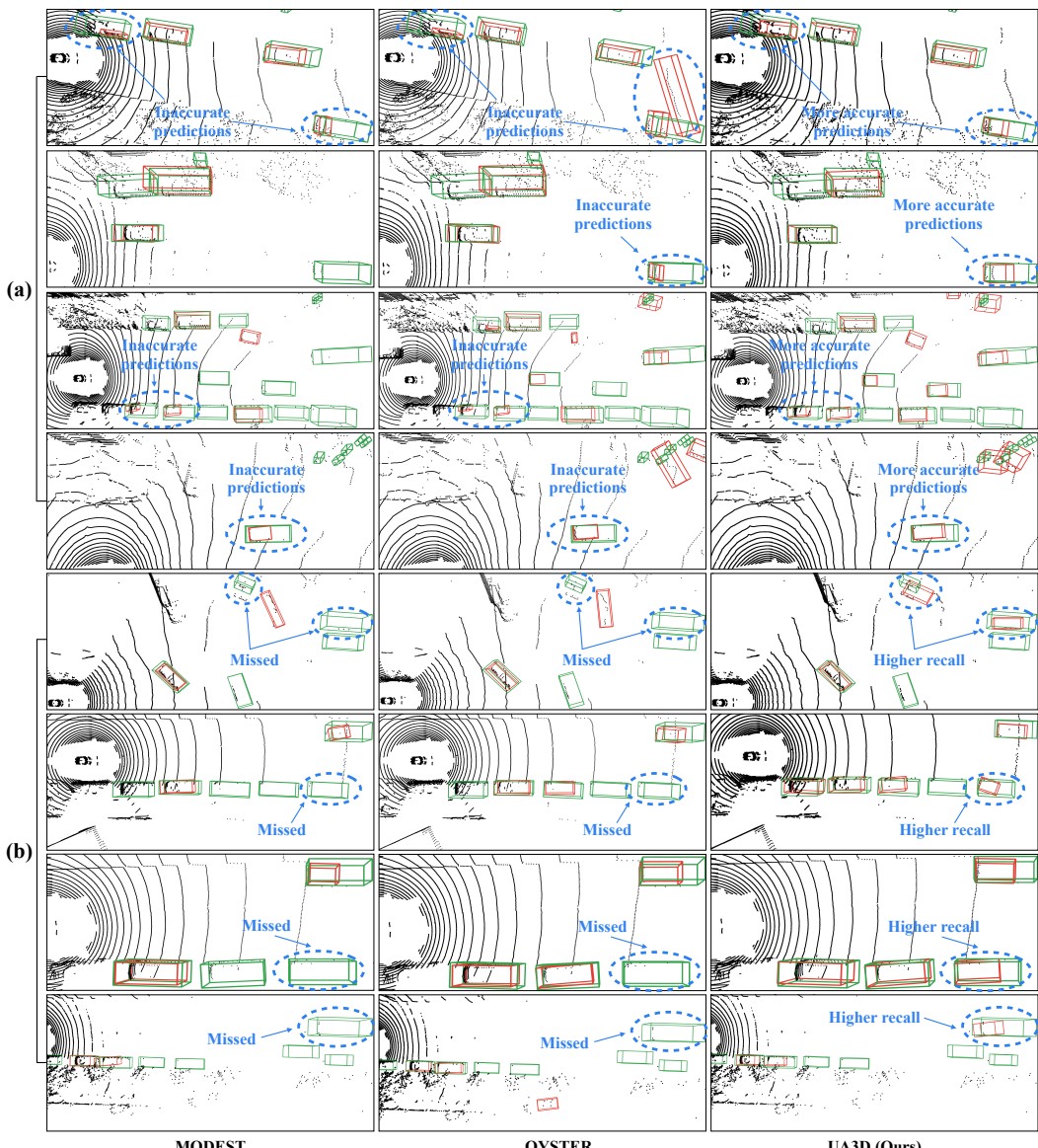

Figure 5: **Further qualitative comparison between different methods.** We compare our uncertainty-aware framework with previous works, *e.g.*, MODEST and OYSTER. **Green** boxes denote the ground-truth and **red** boxes represent predictions from the detection model. (a) Our uncertainty-aware framework shows more accurate perceptions of various foreground objects. (b) In challenging scenarios, such as distant objects with sparse point clouds or small objects, our method achieves a higher recall rate.

This improvement is attributed to our proposed uncertainty estimation and regularization, which mitigate the negative effects of inaccurate pseudo boxes at a fine-grained coordinate level. Fig. 5 (b) further shows that our method is more effective in recalling difficult object categories, *e.g.*, far and small objects. Our uncertainty-aware framework enhances the prominence of accurate pseudo boxes for these challenging objects, facilitating more effective recognition of those objects.

## A.4 EXPLANATION OF UNCERTAINTY VISUALIZATION

We present the explanation of our uncertainty visualization in Fig. 6. The uncertainties in length, width, and height are represented by the gap between the corresponding coordinates of the **purple**

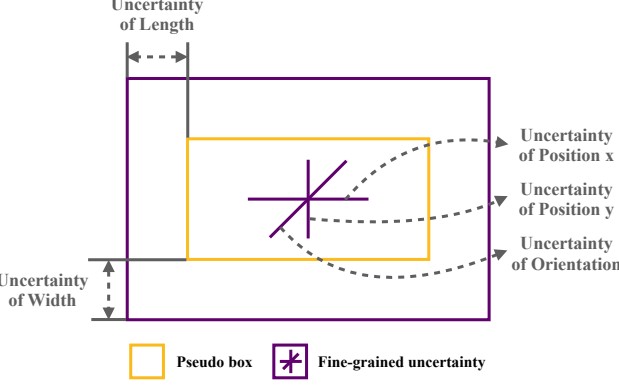

Figure 6: **Detailed explanation of our uncertainty visualization in Bird's Eye View (BEV).** (1) Uncertainty of length: it is visualized by the gap between the length coordinates of the **purple** and **yellow** boxes. (2) Uncertainty of width: it is similarly represented by the gap between the width coordinates of the two boxes. (3) Uncertainty of height: it is depicted as the gap between the height coordinates of the two boxes, though it is omitted in BEV for brevity. (4) Uncertainty of position x: it is shown by the length of the **purple** line extending horizontally (left-to-right). (5) Uncertainty of position y: it is represented by the length of the **purple** line extending vertically (top-to-bottom). (6) Uncertainty of position z: it is visualized by the length of the **purple** line along the z-axis, but it is not shown in BEV for simplicity. (7) Uncertainty of orientation: it is illustrated by the length of the **purple** diagonal line.

Table 4: **Ablation study of loss weight $\mu$ for the auxiliary detector (see Eq. 3).** We observe that a balanced learning process, with equal loss weights for both detectors, produces the best results.

| $\mu$ | 0-30m | 30-50m | 50-80m | 0-80m |
|---|---|---|---|---|
| 0.25 | 33.9 / 22.2 | 5.5 / 2.2 | 2.1 / 0.3 | 15.4 / 8.8 |
| 0.5 | 32.5 / 20.7 | 5.5 / 2.3 | 3.1 / 0.4 | 15.0 / 8.6 |
| 1 | 38.3 / 23.8 | 10.1 / 3.5 | 4.3 / 0.7 | **19.6 / 10.5** |
| 2 | 33.2 / 20.8 | 4.9 / 1.9 | 2.1 / 0.3 | 14.5 / 8.4 |

and **yellow** boxes. For the uncertainties in position (x, y, z) and orientation, they are visualized by the lengths of the **purple** lines along the respective directions.

A.5    FURTHER ABLATION STUDIES

We conduct an ablation study on the loss weight $\mu$ of auxiliary detector (see Table 4). We observe that $\mu = 1$ yields the best detection performance. This suggests that applying equal weights to both branches fosters a balanced learning process, enhancing overall model performance. When the loss weight for the auxiliary detector is reduced to 0.25 or 0.5, our uncertainty-aware framework still outperforms strong baseline (OYSTER), demonstrating the robustness of our approach to variations in hyper-parameters. However, increasing the loss weight to 2 negatively impacts the performance of the primary detector — the one used for final evaluation - likely due to an overemphasis on the auxiliary branch during training.

Additionally, we present an ablation study on the feature extraction backbone layer to which the auxiliary detector is attached (see Table 5). The original feature backbone consists of 4 sa_layers and 4 fp_layers. We refer to those layers as sa_layer_i and fp_layer_i, where i refers to the i*th* layer. We experiment by attaching the auxiliary detector to different layers, *e.g.*, sa_layer_4, fp_layer_1, and fp_layer_2. The auxiliary detection branch mirrors the remaining layers in primary detection branch. For example, when attaching to sa_layer_4, the auxiliary branch contains the same 4 fp_layers as the primary branch. From experiments, we observe that attaching the auxiliary detector to the sa_layer_4 yields the best results. When attaching to the sa_layer_4, we utilize all the FP layers, which facilitates the construction of an independent auxiliary detection branch endowed with full capacity. This maximizes the effectiveness of our proposed uncertainty-aware framework. In contrast, utilizing only 3 FP layers (attaching to fp_layer_1) or 2 FP layers (attaching to fp_layer_2) compromises some feature processing capabilities crucial for 3D detection. Consequently, the auxil-

Table 5: **Ablation study on the specific layer within the feature extraction backbone to which the auxiliary detector is attached.** From shallow to deeper, we study through sa_layer_4, fp_layer_1, and fp_layer_2. We observe that attaching the auxiliary detector to a shallower layer, *e.g.*, the sa_layer_4, yields the best performance.

| Layer | 0-30m | 30-50m | 50-80m | 0-80m |
|---|---|---|---|---|
| sa_layer_4 | 38.3 / 23.8 | 10.1 / 3.5 | 4.3 / 0.7 | **19.6 / 10.5** |
| fp_layer_1 | 34.4 / 21.2 | 9.4 / 3.1 | 4.6 / 0.6 | 18.0 / 9.3 |
| fp_layer_2 | 31.3 / 19.4 | 6.6 / 2.1 | 2.5 / 0.3 | 15.1 / 8.0 |

iary detector tends to produce outputs that are identical to those of the primary detector, diminishing the ability of model to accurately estimate uncertainty.

