# OpenReview forum: "Harnessing Uncertainty-aware Bounding Boxes for Unsupervised 3D Object Detection"
_ICLR.cc/2025/Conference — ICLR 2025 Conference Withdrawn Submission_

### Official Review · Reviewer_9xMD · 2024-11-03

**Soundness:** 2
**Presentation:** 3
**Contribution:** 2
**Rating:** 5
**Confidence:** 4

**Summary:**

This paper introduces an uncertainty-aware framework, UA3D, for unsupervised 3D object detection. It aims to generate different weights for the pseudo bounding boxes by estimating the box uncertainty since the pseudo bounding boxes are usually noisy. In this way, the pseudo bounding boxes can be more reliable. This experiments are conducted on Lyft and nuScenes dataset with a good results.

**Strengths:**

- The paper is well written, and the presentation is clear and easy to follow.
- The proposed methods surpass previous methods such as MODEST and OYSTER.
- Visual results show improvement by the proposed uncertainty estimation.

**Weaknesses:**

- The paper currently lacks a comparison with CPD [1], which uses a similar constrained self-training framework. I strongly recommend the authors to present quantitative results on Waymo or KITTI datasets. Such results are important to substantiating the effectiveness of UA3D through a direct comparison with CPD.
- The paper utilizes a very limited number of samples, 3985 scenes, for training and tests only on 2412 samples from nuScenes datasets. I suggest that the author replace the initial labels method from MODEST with those from CPD or OYSTER, which expand the training and testing samples to cover the entire dataset. This would offer a comprehensive evaluation and demonstrate the scalability of UA3D.
- The significant performance gap between UA3D and fully[2,3]/sparsely[4,5] supervised methods is so large, which makes me concerned about the studies issue or the setting of this issue.
As mentioned in weakness 2, I wish that the authors provide the results on the entire dataset and compare these results with state-of-the-art fully/sparsely supervised methods.
- UA3D is based on PointRCNN which is a method introduced in 2018. Since then, more advanced 3D detectors have been developed. Incorporating state-of-the-art 3D detectors cloud potentially enhance the performance and robustness of the proposed method.
- I personally believe unsupervised 3D object detection has limited practical value. I have never seen an autonomous driving system adopt unsupervised 3D object detection.

[1] Commonsense Prototype for Outdoor Unsupervised 3D Object Detection. CVPR’24 \
[2] DSVT: Dynamic Sparse Voxel Transformer with Rotated Sets. CVPR’23 \
[3] SAFDNet: A Simple and Effective Network for Fully Sparse 3D Object Detection. CVPR’24 \
[4] CoIn: Contrastive Instance Feature Mining for Outdoor 3D Object Detection with Very Limited Annotations. ICCV’23 \
[5] MixSup: Mixed-grained Supervision for Label-efficient LiDAR-based 3D Object Detection. ICLR’24 \

**Questions:**

- The auxiliary detector and the primary detector share the same network architecture and supervision, differing only in the channel dimensions. This raises the question of how these detectors can learn different or distinct information. As an example, in CPD, the two networks differ in point density, which aids in capturing diverse information.
- Would this framework enhance the performance of fully supervised methods if the pseudo labels were replaced with ground-truth labels?

---

### Official Review · Reviewer_5u2P · 2024-11-03

**Soundness:** 3
**Presentation:** 2
**Contribution:** 2
**Rating:** 6
**Confidence:** 3

**Summary:**

This paper proposes a way to do 3D bounding box detection in an unsupervised manner by learning "intelligently" from pseudo-labels. As pseudo-labels are usually noisy, the paper proposes to model the quality of pseudo-labels by taking a vote between two detectors and then adjusting the loss signal based on this agreement / disagreement between the detectors. Experiments show that this is an effective technique which is able to perform better 3D object detection as compared to other unsupervised baselines.

**Strengths:**

- Paper is written well, all possible questions are already rebutted (mixed feelings about this)
- The method is simple and elegant
- There is sufficient comparison to previous approaches and proof that the uncertainty learnt is correlating with the quality of pseudo-labels.
- The benefit to modeling uncertainty the way this paper has is that it is essentially modeling a multivariate distribution of uncertainty (with 7-dimensional error vector) rather than other approaches discussed.

**Weaknesses:**

- I understand that there is enough comparison with other uncertainty based approaches. But what I am not able to understand is that, what kind of pseudo-labels does this uncertainty estimation work with? There is no comparison to differently obtained pseudo-labels, like one way could be clustering points, other ways could be use a zero shot 2D detector on images and back project these detection into 3D using LiDAR points.
- There is also quantitative comparison with only a single kind of backbone which is based on PointRCNN. Does this uncertainty estimation approach translate to other backbones and 3D detectors?

**Questions:**

- I was not clear about how the auxiliary detector is initialized. Is it trained from scratch or fine-tuned from duplicated PointRCNN weights?

---

### Official Review · Reviewer_muKx · 2024-11-04

**Soundness:** 1
**Presentation:** 3
**Contribution:** 2
**Rating:** 3
**Confidence:** 4

**Summary:**

This paper is for unsupervised LiDAR-based 3D object detection called UA3D. The proposed method includes uncertainty estimation and uncertainty regularization. The experiments are conducted on some autonomous driving datasets, nuScenes and Lyft.

**Strengths:**

- The paper is clear, well-written, and easy to follow.
- Experiments are conducted on some large-scale datasets, nuScenes and Lyft.
- The anonymous code and checkpoints are available.

**Weaknesses:**

- The comparison method is only MODEST and OYSTER for the work in 2022 and 2023. There are few experimental comparison methods, and these works were done at least two years ago.
- The baseline is strange for that dense prediction for all points. Two-stage and one-stage object detectors rarely use this manner of bbox regression, which may lead to too low performance for the fully-supervised setting. (Fully supervised performance of nuScenes 22.2/18.2 looks too low.)
- I doubt whether the discrepancy between the two detector predictions can be considered localization uncertainty. At least for some low-quality/low-confidence bboxes may be randomly regressed. Besides, there may be other factors to affect the box regression in two detectors. So more evidence should be provided to support that the discrepancy between the two detector predictions is a good uncertainty representation.
- The method is too hand-crafted and may limited for the PointRCNN detector. Many parameters and thresholds are designed.
- This paper has no discussion of limitations.
- The code and checkpoints are modified after the paper submission deadline on Oct 15, 2024.

**Questions:**

Please provide more qualitative and quantitative results to support that the discrepancy between the two detector predictions can be considered localization uncertainty.

---

### Official Review · Reviewer_s3X4 · 2024-11-04

**Soundness:** 4
**Presentation:** 4
**Contribution:** 3
**Rating:** 3
**Confidence:** 5

**Summary:**

The paper introduces a novel framework, UA3D, aimed at addressing the challenges posed by inaccurate pseudo bounding boxes in unsupervised 3D object detection tasks. The key contributions of the paper are as follows:

1. The authors propose a method to estimate the uncertainty of pseudo bounding boxes at a fine-grained coordinate level. This is achieved by incorporating an auxiliary detection branch alongside the primary detector within the model, allowing for the assessment of prediction discrepancies that serve as indicators of uncertainty.

2. Based on the estimated uncertainty, the framework adaptively adjusts the loss weights of different pseudo box coordinates during the iterative training process. This approach reduces the influence of inaccurate pseudo boxes on the training, thereby mitigating their negative impact.

**Strengths:**

First and foremost, I appreciate the authors for providing the anonymous link to their code, although I have not been able to test it personally. The paper meticulously delineates the challenge of inaccurate pseudo boxes in unsupervised 3D object detection and adeptly outlines how their proposed framework tackles this problem. The methodology is articulated with clarity, particularly in the sections detailing the uncertainty estimation and uncertainty regularization phases, which greatly facilitates comprehension for the readers.

**Weaknesses:**

1. While the paper introduces a new uncertainty-aware framework to mitigate the impact of noisy pseudo bounding boxes, it lacks a detailed exploration of the model's robustness under various conditions. It would be insightful to see how the model performs under different noise levels, in various weather conditions, and against adversarial attacks to truly assess its robustness.

2. The paper mentions the performance improvement compared with existing methods, but does not compare the differences between different methods in dealing with pseudo bounding box noise in detail. It is recommended to provide a detailed comparison with existing methods (such as the UNION [1] method and the CPD [2] framework), including methodology, performance, and advantages and disadvantages analysis.

3. The paper mentions the performance improvement of the model on the nuScenes and Lyft datasets, but does not mention the performance differences of the model on different types of objects (such as pedestrians, vehicles, cyclists, etc.). It is recommended to conduct a detailed analysis of the performance of the model on different types of objects and explore how to improve the model to improve the detection performance of minority categories.

Reference

[1] Lentsch, T., Caesar, H., & Gavrila, D. M. (2024). UNION: Unsupervised 3D Object Detection using Object Appearance-based Pseudo-Classes. arXiv preprint arXiv:2405.15688.

[2] Wu, H., Zhao, S., Huang, X., Wen, C., Li, X., & Wang, C. (2024). Commonsense Prototype for Outdoor Unsupervised 3D Object Detection. In Proceedings of the IEEE/CVF Conference on Computer Vision and Pattern Recognition (pp. 14968-14977).

**Questions:**

See the weakness.

---

### Note · Authors · 2024-11-15

I have read and agree with the venue's withdrawal policy on behalf of myself and my co-authors.